# Endothelial Cells and Mitochondria: Two Key Players in Liver Transplantation

**DOI:** 10.3390/ijms241210091

**Published:** 2023-06-13

**Authors:** Alessandro Parente, Mauricio Flores Carvalho, Andrea Schlegel

**Affiliations:** 1HPB and Transplant Unit, Department of Surgical Science, University of Rome Tor Vergata, 00133 Rome, Italy; aleparen@gmail.com; 2Division of Hepatobiliary and Liver Transplantation, Department of Surgery, Asan Medical Center, University of Ulsan College of Medicine, Seoul 05505, Republic of Korea; 3Department of Experimental and Clinical Medicine, University of Florence, 50121 Florence, Italy; drmauras@gmail.com; 4Fondazione IRCCS Ca’ Granda, Ospedale Maggiore Policlinico, Centre of Preclinical Research, 20122 Milan, Italy; 5Transplantation Center, Digestive Disease and Surgery Institute, Department of Immunity and Inflammation, Lerner Research Institute, Cleveland Clinic, Cleveland, OH 44106, USA

**Keywords:** mitochondria, reactive oxygen species, ischemia-reperfusion injury, endothelial cells, shears stress, machine perfusion

## Abstract

Building the inner layer of our blood vessels, the endothelium forms an important line communicating with deeper parenchymal cells in our organs. Previously considered passive, endothelial cells are increasingly recognized as key players in intercellular crosstalk, vascular homeostasis, and blood fluidity. Comparable to other cells, their metabolic function strongly depends on mitochondrial health, and the response to flow changes observed in endothelial cells is linked to their mitochondrial metabolism. Despite the direct impact of new dynamic preservation concepts in organ transplantation, the impact of different perfusion conditions on sinusoidal endothelial cells is not yet explored well enough. This article therefore describes the key role of liver sinusoidal endothelial cells (LSECs) together with their mitochondrial function in the context of liver transplantation. The currently available ex situ machine perfusion strategies are described with their effect on LSEC health. Specific perfusion conditions, including perfusion pressure, duration, and perfusate oxygenation are critically discussed considering the metabolic function and integrity of liver endothelial cells and their mitochondria.

## 1. Introduction

Endothelial cells (ECs) build the inner layer of our blood vessels and represent an important barrier, communicating with cells in deeper tissues, thereby maintaining vascular homeostasis in our body [1]. While ECs were often considered “passive”, such cells are increasingly recognized today as key players in intercellular crosstalk [2]. Multiple dynamic blood flow patterns affect the luminal surface of endothelial cells, also based on the link between shear stress and immuno-metabolic functions imposing different cellular phenotypes [3]. Although the role of optimal mitochondrial respiration (OXPHOS function), which protects endothelial cells from acute and chronic injury, is increasingly understood, most studies target cardiovascular endothelial cells [1]. Despite the known key role of healthy sinusoidal endothelial cells (SEC) for liver function maintenance in acute and chronic liver injury and the development of cirrhosis, the link between mitochondrial metabolism and SEC protection during ischemia-reperfusion injury (IRI) is less well understood. Liver SECs can remain quiescent for decades and suddenly change to highly activated status by responding to the release of reactive oxygen species (ROS), danger-associated molecular patterns (DAMPs), cytokines and other molecules, that stimulate growth and repair induced by vascular endothelial growth factor (VEGF) signaling. Recently, the increasing use of machine perfusion for the optimization of liver graft preservation in the setting of transplantation has led to a better understanding of the role of mitochondria and SECs.

This article describes the underlying mechanisms of SEC injury and protection during different IRI phases in the context of liver transplantation. The role of flow dynamics, shear stress, and related processes are also specifically discussed in the context of future therapeutic interventions. The currently tested ex situ perfusion techniques are discussed next, together with their impact on mitochondrial function and potential to protect liver SECs, serving as important measures to prevent severe acute and ongoing inflammation of the liver microenvironment after transplantation.

## 2. Mitochondria and Endothelial Cells under Physiological Conditions

Mitochondria are increasingly recognized for their fundamental role in all cells and processes of health and disease, including aging and cancer development, but also immune system activation and recipient outcomes after solid organ transplantation [4,5,6,7].

The continuous energy production (i.e., Adenosine-trisphosphate: ATP) and balance maintenance between the production and removal of ROS are two fundamental roles of mitochondria. Of particular importance in endothelial cells is the contribution to required cellular and mitochondrial calcium levels. The proper mitochondrial function does strictly depend on the mitochondrial membrane potential, generated by complexes I, III, and IV providing the full OXPHOS capacity to produce ATP. Despite this well-described metabolism, studies demonstrate diverse metabolic profiles and phenotypes of endothelial cells in different organs, which relate to the specific position in the vessel, based on more or less exposure to flow and shear stress. Endothelial cell metabolism, such as fatty acid oxidation (FAO) or glycolysis, triggers vessel proliferation together with genetic signaling for growth factor expression [8]. The various metabolic features observed in endothelial cells are increasingly explored to develop novel therapeutic strategies [8].

The plasticity observed in the metabolic expression of endothelial cells as a response to physiological changes is truly remarkable [9]. For example, while renal ECs were found to impressively upregulate their ATP production as a response to water deprivation [10], pro-angiogenic ECs depend on glycolysis in addition to OXPHOS [9]. Of interest is also the upregulation of glycolysis during IRI in cardiac endothelial cells [11].

The EC plasticity also depends on the type of vessel involved. Li et al. recently demonstrated the dominance of OXPHOS in arterial ECs compared to more pronounced glycolysis occurring with lower oxygen levels in cells located in veins or arteriovenous axes [2,9]. Mitochondria and their energy production are therefore more important for EC function maintenance than previously appreciated.

The various blood flow patterns play an additional key role and may trigger a metabolic switch in ECs. Hong et al. recently explored such features in a model of carotid artery ligation. The authors demonstrated that mitochondrial fragmentation is increased in ECs exposed to disturbed flows. The significant induction of mitochondrial fragmentation was associated with EC activation. In contrast, elongated mitochondria were predominant in ECs exposed to unidirectional flow [12], a flow pattern found to decrease mitochondrial fragmentation and improve fatty acid uptake and OXPHOS [1]. Conversely, disturbed flows trigger EC activation and switch to pro-inflammatory phenotypes with increased glycolysis and reduced OXPHOS [9,13]. Based on such recent studies, the augmentation of OXPHOS might be a promising target for new therapeutics to reduce the acute inflammatory response and potentially prevent ongoing inflammation.

Next, mitochondrial homeostasis and remodeling are further key features to maintain EC function. The related processes of mitochondrial quality control cover the fine balance between fission and fusion [9,14]. While standard levels of fission are essential to maintain mitochondria healthy, it also appears crucial to control mitochondrial DNA and mitophagy and intracellular distribution. As seen with all metabolic functions, excessive and prolonged mitochondrial fission is related to cellular dysfunction, reducing OXPHOS with elevated mitochondrial ROS release (mtROS) and the development of cardiovascular diseases. Of interest is that well-known risk factors, including smoking or diabetes and higher levels of proinflammatory cytokines (i.e., TNF-alpha), trigger such elevated fission contributing to the development of cardiovascular diseases [9,15].

The large form of cytosolic GTPase (Drp1) acts as a key player during mitochondrial fission. Having various sites for phosphorylation, the molecule Drp1 can trigger different levels of GTPase activity. Being translocated to the outer mitochondrial membrane, Drp1 causes the GTP-dependent production of a “daughter” mitochondrion through the constriction of otherwise tubular mitochondria with a newly synthesized ring.

The impact of different physiological flow patterns on mitochondrial metabolism was recently assessed in animal models and systems of microfluid circulation. The aim of this study was to critically analyze immunophenotypic features of ECs exposed to specific flow patterns (i.e., unidirectional vs. disturbed), triggering morphological changes in EC mitochondria. Such underlying mechanisms are largely unknown under physiological conditions in the main cardio-vasculature and even more with injury, such as hypoxia and IRI in specific organs. We may well expect such protective ECs features to change rapidly when machine perfusion technology is applied to organs ex situ, lacking communication with the entire body, and moreover, when “treating” isolated organs with various unphysiological perfusates with more or fewer additives and at all temperatures [16,17].

## 3. Role of Mitochondria and Endothelial Cells during Ischemia-Reperfusion Injury

Various complex biochemical processes are involved in the biphasic phenomenon of IRI. During ischemia (lack of oxygen), healthy phenotypes of liver cells become deregulated. While the role of mitochondria as instigators of IRI was primarily discussed in dominant liver cells, such as hepatocytes or macrophages as links to activate the circulating recipient immune cells, the important role of SECs is being increasingly recognized. The role of mitochondrial respiration in SECs was rather neglected based on the relevance of glycolysis for energy production [9]. However recent studies demonstrated that endothelial cell mitochondria are highly active and equally relevant for biosynthetic and signaling functions, which depend on the capability to produce enough ATP [8,18].

While mitochondrial dysfunction and injury are often assessed in hepatocytes, macrophages (Kupffer cells), and neutrophils, the role of oxidative endothelial cell injury is also less well described due to the complex interplay of various cells found in the microvascular environment of liver sinusoids. Albeit the importance of mitochondrial complex I–V dys- or non-function during ischemia with the described lack of ATP is described in various liver cells (Figure 1), the understanding of mitochondria’s key role in SECs is lagging behind.

Sinusoidal endothelial cells, with their role as the first liver sinusoidal barrier, are particularly vulnerable to IRI injury and become activated through various trigger molecules either from their own mitochondria or as a response to signals from neighboring cells including hepatocytes, Kupffer cells, or recipient blood cells after transplantation. Liver SECs are localized at the interface between (recipient) blood cells on one side and hepatocytes, Kupffer cells, and stellate cells on the liver tissue side. Severe oxidative SEC injury causes edema, hypoperfusion with secondary hypoxia due to a disruption of the microcirculation, and tissue necrosis [19].

Although a significant part of these mechanisms remains unknown, the major instigators of this proinflammatory IRI cascade are liver mitochondria. Mitochondrial respiration switches to an anaerobic status when oxygen is lacking. Respiratory chain complex proteins reduce and stop their electron flow with the result of energy depletion and a lack of ATP. This is paralleled by a dys- and/or non-function of the TCA cycle with the accumulation of succinate and other potentially toxic metabolites [20,21,22,23]. Mammalian cells upregulate their defense mechanisms with an increase in cellular itaconate levels under hypoxic conditions at 10 degrees [24,25]. 

Another important consequence, particularly for SECs, is seen with the oxidation of reduced glutathione (GSH) and NAD(P)H. These processes increase mitochondrial Calcium levels and lead to pH changes with the direct loss of the mitochondrial membrane potential due to a collapse of inner mitochondrial membrane ion gradients. This mitochondrial membrane permeability transition is also characterized by mitochondrial depolarization and OXPHOS uncoupling with further ATP depletion [19,26,27]. The membrane potential collapse in turn blocks the Calcium uptake by mitochondria and enhances ROS production [26,28,29].

Upon the reintroduction of oxygen at reperfusion or transplantation, mitochondria aim to immediately resolve the energy depletion, thus electrons pass though complex proteins in an undirected way, thereby initiating the release of ROS from complex I. Of note, this occurs within the first few minutes after reoxygenation, initiating the unstoppable IRI cascade with the downstream release of mitochondrial DNA, DAMPs, and cytokines. Historically, this phenomenon of ROS release was predominantly allocated to liver macrophages [30]. However, today there is a general understanding that the mitochondrial trigger occurs in all liver cells, which is a particularly critical event during reperfusion in liver SECs [19]. The initial ROS release may also activate Kupffer cells, which express higher levels of selectins and adhesion molecules to attract recipient neutrophils attaching to liver SECs after transplantation.

Once initiated, the IRI triggered by hepatocytes may trigger another wave of inflammation seen in cells that survived the initial IRI without major injury. However, such cells may release new ROS molecules attracting recipient blood cells (i.e., neutrophils and platelets) to attach to liver sinusoids or migrate through this barrier with secondary hypoxia and ongoing severe IRI (Figure 2) [31]. This may lead to sinusoidal obstruction with secondary hypoxia and ongoing IRI features in the liver periphery. The direct result is liver edema, hypoperfusion, and necrosis. The best possible understanding of the underlying mechanisms of IRI in liver SECs is therefore of the utmost importance to design tailored interventions.

Despite the increasing understanding of how such metabolic processes impact, certain levels of ROS were found to activate protective mechanisms in liver SECs including autophagy, particularly during the early phase of IRI [32]. Various autophagy-related proteins (ATGs) were described to regulate this inherent, evolutionary process [27]. Based on sequential maturation steps, double-membrane structures (autophagosomes) are built to cargo damaged, dysfunctional, or potentially threatening subcellular or foreign structures, such as mitochondria (mitophagy), endoplasmatic reticulum (ERphagy), or bacteria (xenophagy) [27]. The formed autophagosomes then merge with lysosomes to degrade the “enclosed” cargo. Interestingly, such substrates released from the cargo are used as a source for anabolic processes. Autophagy is a protective mechanism traditionally explored in SECs. Recent studies demonstrate the role of Kruppel-like-factor 2 (KLF2) as the main regulator, a phenomenon previously described in organs, other than the liver (Figure 2) [33,34]. Such studies support the hypothesis that failure to initiate autophagy can lead to SEC death as a response to oxidative stress and IRI inflammation. Futile levels of IRI with autophagy failure are associated with microvascular dysfunction [27,33,34].

## 4. The Role of Endothelial Shear Stress

Modern research confirms that severe mitochondrial dysfunction triggers SEC activation and apoptosis. To identify effective interventions, it is imperative to assess how mitochondria in SECs behave under various physiological and pathophysiological conditions such as different blood flow patterns [9].

Blood-flow-initiated laminar shear stress (LSS) is described in all vascular ECs, where inherent protective mechanisms are mediated by direct upregulation of KLF-2, triggering the transcription of anti-oxidant, anti-thrombotic, and anti-inflammatory genes of this main response to certain LSS levels [35,36,37,38]. ECs and blood vessels are stabilized, maintaining their stiffness and protecting them from ongoing inflammation, atherosclerosis, and other diseases [39]. The consistent lack of KLF-2 would lead to cardiac failure after long-term high output responding to a too-low peripheral resistance as recently shown in a model of KLF-2 knockout mice [39,40]. KLF-2 is the key mediator when adult ECs face inflammatory challenges. During ex situ kidney and pancreas perfusion, the pressures are usually chosen with rather high values of up to 25 mmHg or even more to maintain enough shear stress [41,42].

In contrast, too-low or disturbed shear stress (DSS) leads to only weak induction of KLF-2 in exposed ECs, thereby supporting the development of atherosclerosis in such vessels [43]. Various conditions further exacerbate the atherosclerosis risk, including hyperglycemia and high levels of low-density lipoprotein (LDL), two conditions where KLF-2 expression in endothelial cells is reduced [44].

Despite an increasing number of studies focusing on this topic during the past decade, specific details on how KLF-2-expression is induced by different blood flow remain unclear and not explored well enough in livers.

Compared to other capillaries, liver sinusoids have rather low shear flows to enable the contact and exchange of molecules and fluid with surrounding cells [32,45]. Similarly, as seen in other small-diameter vessels, responding to elevated shear stress, liver SECs produce vasodilators and regulate the blood pressure in the hepatic periphery [45]. As described above for other ECs, this event is mediated by the transcription of KLF2 with the subsequent release of nitric oxide (NO) through the upregulation of endothelial nitric oxide synthase (eNOS) activity [32]. It is believed that liver SECs regulate sinusoidal blood flow through hepatic stellate cells (HSC) in the space of Disse. These cells were found to express molecules, such as smooth-muscle actin (alpha-SMA) and desmin, which enfold liver sinusoids. Shear stress leads to a reduction of vasoconstrictive factors including endothelin 1 (ET-1), a mechanism mediated by KLF2. To further support the overall vasodilatation, such molecules act on HSC to avoid vasoconstriction. With an upregulation of ET-1 and KLF2, HSCs maintain their dormant state [32,45]. The proximal position of liver SECs and HSCs enables various interactions among these cell types through growth factors and other mediators (i.e., platelet- and stromal-derived growth factors, CXCR4, CXCL12, and SDF1alpha) [33,34,45].

Another important key feature of SECs appears with the enormous endocytic capacity and the expression of receptors to scavenge molecules. Such features permit the high clearance activity of SECs through the recognition and internalization of extracellular ligands to clean liver sinusoids from waste products and other toxins. Despite the barrier function, these characteristics together with their discontinuous lining enable SECs to freely exchange molecules between sinusoidal blood and parenchymal cells [46,47]. Of interest are also sinusoidal fenestrae, dynamic structures formed through the fusion of opposite membranes (luminal and abluminal). Actin filaments and microtubules connect fenestrae with the cytoskeleton (Figure 3) [45,46]. This mesh-like construction further supports the endocytic molecule transfer between blood and parenchyma [46,47]. Liver SECs can express certain receptors, for example, Fcy, which triggers the clearance of immune molecules such as circulating IgG [48].

Based on their key role in the innate and adaptive immune response, liver SECs are the focus of a variety of research topics in solid organ transplantation. SECs become stimulated by various toll-like receptors expressed on other parenchymal- and non-parenchymal cells as a response to ROS, DAMPs, and other pro-inflammatory molecules.

Hypoxia-triggered mitochondrial injury and dysfunction are linked to immune response through the entire IRI cascade [4,49,50,51,52]. Underlying key mechanisms of shear-stress-mediated oxidative mitochondrial injury should therefore be evaluated further together with compensatory changes in mitochondrial dynamics (i.e., mitophagy and clearance of severely injured mitochondria) and the antioxidant response (i.e., NRF-2 upregulation (Figure 3)). Together, such mechanisms are thought to suppress ROS-mediated inflammation triggered by disturbed flows.

Although these concepts are challenging to explore in studies, they are the perfect example of hormesis. Mild EC stress leads to limited KLF-2 pathway upregulation initiating the release of some ROS with adaptive responses that protect the entire organism. In contrast, severe shear stresses may overwhelm the system or “overrun” the adaptive mechanisms. Such disturbed shear stress can induce high ROS levels, thereby limiting the protective anti-inflammatory responses. The ROS-mediated gene upregulation contributing to a higher sensitivity towards other risk factors and general vascular diseases appears to be a direct consequence [35].

In liver SECs that are positioned in vessel regions with exposure to more DSS, mitochondrial fragmentation is increased. Such mechanisms are initiated by higher expression of dynamin-related protein 1 (Drp1). Further consequences were described above and included mtROS release, a metabolic switch to glycolysis, hypoxia-inducible factor (HIF-1alpha) release, and finally, EC activation [1,9]. Various experimental studies showed that Drp1 inhibition may serve as an interesting strategy to attenuate these proinflammatory mechanisms [1]. The Drp1 enhanced mitochondrial fragmentation in experimental studies using models of carotid artery ligation and microfluid circuits [1,9]. Such findings were further paralleled by increased OXPHOS and fatty acid uptake pointing to a well-functioning mitochondrial complex I through exercise or unidirectional flow (high shear stress) (Figure 3).

More studies are required to detail such processes in the liver in both physiological and pathological situations [35]. To better understand and address the IRI-associated microvascular inflammation after liver transplantation, the complex interplay between mitochondria-initiated injury (in all liver cells) that occurs directly after reoxygenation should be explored together with the secondary mitochondrial response to shear stress with subsequent ROS release.

## 5. Effect of Machine Perfusion on Mitochondria and Endothelial Cells

To overcome the low utilization rates and enable the safe use of marginal organs, dynamic perfusion techniques are increasingly tested in clinical trials and implemented in routine practice in a few centers worldwide [53]. All techniques have the aim to reduce ischemia with the re-introduction of oxygen either replacing cold storage or in addition to cold storage after organ transport. The microvascular environment is a key factor for improved oxygen distribution and perfusion quality. Required perfusion flow, pressure, and resistance are therefore frequently discussed in the context of the two main ex situ perfusion techniques. A red-blood-cell-based perfusate is used for normothermic machine perfusion (NMP) at physiological temperatures of 37 °C. Both liver inflow vessels are cannulated, and the safety and feasibility were demonstrated together with a reduction of early allograft dysfunction (EAD) in two large randomized controlled trials (RCTs) using standard livers [54,55]. With advanced donor risk and additional cold storage of 6–7 h prior to NMP, outcomes were comparable to standard cold storage alone [56,57]. Known as hypothermic oxygenated perfusion (HOPE), the second main type of ex situ perfusion technique introduces oxygen at cold conditions (4–12 °C) [58]. Four RCTs are published with livers from extended criteria donors (ECD), demonstrating lower EAD rates, the reduction of complications, and better graft survival [59,60,61]. This technique is routinely performed after a median of 6 h cold storage [62]. Subnormothermic machine perfusion (SNMP) is the third technique, applied at 16–25 °C, particularly in the context of endothelial cells, which are sensitive to cold ischemia. However, no large RCTs are available with SNMP yet [63,64]. Although most perfusion conditions are fairly well established and uniformly applied, key parameters such as pressure and related flows are based on the physiological situation in the human body at 37 °C. The following chapters summarize and discuss the currently available literature, focusing on the effect of ex situ perfusion techniques on ECs and mitochondria.

### 5.1. Hypothermic Oxygenated Perfusion (HOPE)

There is increasing evidence for the protective effect of a short period of hypothermic oxygenated perfusion (HOPE) before liver transplantation [58]. Based on an ever-increasing body of experimental studies, fairly homogenous perfusion conditions and timings are currently established and used throughout Europe and a few other selected countries [62]. Following standard organ procurement, cold storage, and organ transport, livers undergo hypothermic perfusion with an artificial solution (i.e., Belzer MPS) that is highly oxygenated with a pO_2_ between 60 and 100 kPa [65]. This cold perfusion (4–12 °C) is performed after cold storage, i.e., during recipient hepatectomy for a median of 2 h [60,61,62,66,67]. Two main concepts exist for this perfusion technique, with the first one where only the portal vein (PV) is cannulated with a median pressure of 3–5 mmHg to avoid additional endothelial injury. The dual-HOPE concept (D-HOPE) involves the additional cannulation of the hepatic artery (HA) (ideally proximal using the aortic patch, the truncus or the entire aorta) applying a pressure of 25 mmHg [66]. More research is needed to define the best possible perfusion parameters under hypothermic conditions for livers with different donor risk profiles and to achieve the required mitochondrial reprogramming [54]. Most perfusion parameters are kept fairly uniform during hypothermic perfusion, including temperature, high perfusate oxygen levels, and duration.

The two most important parameters in the context of mitochondria and endothelial cells, perfusate oxygen concentration and perfusion pressure (flow), are discussed below with the background of perfusion temperature and duration [65,68].

#### 5.1.1. Hypothermic Machine Perfusion and Mitochondria

When oxygen is reintroduced, mitochondria release ROS immediately in all cells, related to the initial quality and how well the cells could handle the injury accumulated in the donor and during procurement and preservation. Such ROS release was described to be temperature dependent with higher levels under warm conditions when compared to cold temperatures [69,70]. Cells severely affected by IRI will die with subsequent molecule release (i.e., DAMPs and cytokines), which in turn injure and activate neighboring cells, thereby transforming the tissue into a status of ongoing inflammation. With a significantly higher number of hepatocytes and more than 2000 mitochondria in each cell, this cell population is largely responsible for the initial hit at liver reperfusion. However, the entire metabolism of all other cell types strongly depends on the function of mitochondria.

Research in different organs, for example in kidneys, strongly suggests that high levels of perfusate oxygen are needed to achieve such a metabolic switch with ATP reloading in mitochondria [71]. Lazeyras et al. demonstrated the need for a perfusate pO_2_ of almost 100 kPa to completely recharge cellular ATP as confirmed through magnetic resonance spectroscopy in tissues obtained from kidneys during and after hypothermic oxygenated perfusion (HOPE) [71]. With the known role of endothelial cells, the structure with the first contact with perfusion fluid and oxygen, such cells should benefit most from HOPE perfusion, given perfusion pressures are well controlled.

The findings by Lazeyras et al. were paralleled in experimental studies with other solid organs. The HOPE approach was either performed with different levels of oxygen or the oxygen molecule was entirely removed from perfusates using nitrogen. The group from Bonn assessed the role of oxygen during cold liver perfusion with three different perfusate oxygen levels, i.e., 0%, 20% (equal room air), and 100% perfusate oxygenation [72]. Of note, livers undergoing HOPE with proper oxygenation achieved higher levels of protection, more energy reloading, and significantly better functional recovery as demonstrated by more bile production [72,73].

Most physiological adaptations and treatment effects were assessed under physiological conditions and with healthy ECs. Similarly, many changes imposed by hypothermic oxygenation remain unknown. Only recently were some of the protective effects of HOPE resulting in mitochondrial metabolism changes described. Though most cells have the same metabolic features during inflammation and survival, mechanisms require confirmation in other cell types, such as ECs.

#### 5.1.2. The Effect of Hypothermic Machine Perfusion on Liver Endothelial Cells

The described protection of other cells through HOPE, i.e., hepatocytes and Kupffer cells via mitochondrial reprogramming and lower levels of ROS, DAMPs, and cytokine release after later transplantation, also has indirect effects on SECs. The reduced tissue acute and ongoing inflammation in sinusoids leads to better peripheral liver perfusion with only limited or no hypoxia secondary to sinusoidal obstruction through massive platelet and neutrophil attachment and migration.

Next, the above-described protective effects of cold oxygenation on mitochondrial metabolism are also expected in LSECs, although the number of studies is limited. A few groups have explored the impact of hypothermic liver perfusion on liver SECs often in a descriptive design and with a focus on perfusion duration and pressure in relation to sheer stress [74].

A total of ten studies tested the effect of HOPE or D-HOPE on endothelial cells (Table 1). Six studies used an animal model (either porcine or rodent), and one tested such effects in discarded human livers. Tissue samples from two clinical transplant studies were assessed for EC protection with HOPE and correlated with posttransplant outcomes [75,76]. One study combined rodent experiments with human liver transplants in a model of severe macrosteatosis [77].

The Zurich group presented the effects of HOPE after 12 h of cold storage of extended criteria donor grafts (ECD) with advanced macrosteatosis. Rodent livers with at least 60% macrosteatosis were protected with a lower expression of MMP3 and 8, TIMP, and CXCL10 after HOPE [77]. Recipients transplanted with such HOPE-treated macrosteatotic grafts had better survival rates and reduced liver injury. Such findings were paralleled by similar results with clinical transplantation of steatotic human livers, both from donation after circulatory death (DCD) and brain death (DBD). Recipients of such organs demonstrated better early liver function and improved survivals with lower retransplantation rates after HOPE, which was in contrast to cold storage controls [77]. The same study demonstrated a better sinusoidal perfusion quality assessed with fluoresceine during early reperfusion after transplantation of HOPE compared to cold-stored control grafts. Such findings show the HOPE effect on the sinusoidal space. In 2014, Op den Dries et al. assessed 18 porcine DCD livers during normothermic reperfusion. Lower signs of arteriolonecrosis of the peribiliary vascular plexus and liver SECs were protected as demonstrated by lower caspase 3 positivity after D-HOPE, compared to cold storage alone [78].

Kanazawa et al. explored the effect of D-HOPE in porcine livers after circulatory death. This study demonstrated less inflammation and lower liver sinusoidal obstruction through a reduced number of aggregating platelets on SECs. Paralleling various other studies, the authors demonstrated ATP recharging with D-HOPE [79]. Despite the fact that this model did not include liver transplantation but only normothermic reperfusion, the results are of importance because they demonstrated the HOPE effect despite a high injury of 60 min donor warm ischemia time [79].

The Groningen group presented another study with discarded human livers. Burlage et al. explored the effect of D-HOPE on liver SEC integrity and dysfunction [80]. An overall number of 15 DCD livers and 3 controls (DBD) underwent 2 h D-HOPE followed by 6 h NMP. Various surrogate markers for EC function and injury were assessed during this time. Compared to cold-stored DCD livers and healthy controls (DBD livers), DCD livers, which underwent additional endischemic D-HOPE, demonstrated a different gene profile. Genes encoding key mediators, including KLF2, TM, eNOS, HIF-2α, VEGFa, and HO-1, were upregulated and highly expressed during NMP after previous D-HOPE. At the end of six hours of NMP, the expression of TM, eNOS, HIF-2α, and VEGFa was significantly higher after D-HOPE compared to the static cold storage (SCS) alone [80]. The eNOS and TM expression, for example, were 9- and 6-fold higher after D-HOPE compared to the controls, respectively [80]. The key mediator for HSC contraction and functional narrowing of the SECs, ET-1, was lower with D-HOPE. The subsequent effect was better endothelial cell function observed through the upregulation of mechano-sensitive cytoprotective genes, with better perfusion flows in both inflow vessels [80]. The authors also showed restored endothelial cell viability of ECD livers (both DCD and DBD) after D-HOPE as demonstrated by lower microvascular injury using the histological scoring system at the end of subsequent normothermic reperfusion [80]. Regarding mitochondrial metabolisms and liver SEC injury, no differences were observed comparing HOPE and D-HOPE in another experimental study [74]. The function and injury of ECs were assessed by perfusate NO, antigen levels of Willebrand factor, and gene expressions including eNOS, ET-1, KLF-2, CD31, vWF, and VEGFa (Table 1). This body of experimental studies was supported by two clinical trials.

In 2012, Guarrera et al. found a significantly reduced proinflammatory cytokine expression and lower expression of adhesion molecules and neutrophils and macrophages migration after HMP compared to cold-stored controls [76]. This study was recently paralleled with additional clinical samples by a study from Italy. The team from Bologna has recently introduced the HOPE concept in their clinical practice and confirmed this with their RCT published in 2022 [60]. Tissue samples from this study were assessed for EC integrity and activation. Vasuri et al. analyzed 47 human livers from a randomized controlled trial (RCT) in Italy transplanting ECD livers with HOPE or cold storage [60,75]. The authors describe endothelial trophism and intracellular homeostasis measured through Nestin expression after HOPE. Additionally, liver SEC endothelization is an interesting parameter useful to assess the response to elevated portal pressures that can show a metaplastic EC shift to a cell subtype with reduced specialization being protected from high portal pressure [75].

#### 5.1.3. The Role of Perfusion Pressure, Shear Stress, and Flow

Clinical HOPE perfusions are either performed through the PV only or in a combined mode through both liver inflow vessels. Perfusion parameters used for clinical machine perfusion are often extrapolated from the physiological situation in human organs at 37 °C. Although most perfusion systems are pressure controlled, the applied conditions should provide enough unidirectional perfusion flow to trigger the required shear stress protecting mitochondria and endothelial cells and achieve the complete distribution of oxygen, while at the same time, too-high perfusion pressures that injure liver SECs should be avoided [81].

T’Hart et al. explored these phenomena in a systematic analysis using different perfusion pressures [81]. Portal vein pressures of 2, 4, or 8 mmHg were combined with pressures of 12.5, 25, and 50 mmHg in the HA. The best results were seen with a PV and HA pressure of 4 and 25 mmHg, respectively. With such conditions, good cortical perfusion, minimal endothelial cell death, and ATP reloading were achieved. Higher pressures triggered more liver cell injury, while too-low pressures resulted in hypoperfusion [81]. Although this is a relevant study, the authors used only three livers with limited injury (i.e., from braindead donors) per study group and the experiment with PV perfusion only is lacking. This has been addressed by the team from Zurich in an experimental study on perfusion conditions with HOPE of porcine DCD livers (60 min warm ischemia). Organs underwent one hour of endischemic HOPE through the PV with either 3 mmHg or 8 mmHg, paralleling the results from T’Hart et al. [82]. The authors described the protective effect of all liver cells with HOPE at 3 mmHg. Liver SECs were better protected from activation and inflammation at low perfusion pressures. Additional effects were described on mitochondria. High perfusate oxygen levels protected mitochondria uploaded ATP and reestablished complex I function as measured by NADH metabolism. Another interesting finding was that high and low perfusate oxygen levels were equally protective for liver SECs, provided that the PV pressure did not exceed 3 mmHg (Table 1) [83]. Of interest is also the perfusion flow subsequent to a pressure of 3 or 8 mmHg during HOPE; flow rates of 0.05 mL/g liver/min (63–65 mL/min) and 0.12–0.13 mL/g liver/min (150–160 mL/min) were observed, respectively [82]. Another study from Zurich on DCD rodent models confirmed the SEC protection through HOPE [83].

Based on the available experimental studies, the majority of clinical HOPE and D-HOPE perfusions are performed with a PV and HA pressure of 3–5 mmHg and 25 mmHg, respectively [53,62,66]. Despite the impact of various confounders, such values appear to be safe for liver SECs in the context of the currently not-too-extended perfusion duration. Perfusion pressure, flow, and resistance also depend on the temperature. The perfusion at 4 °C was found to induce vasoconstriction and impair liver SECs resulting in increased vascular resistance and morphological changes [84,85]. With exposure to additional high perfusion pressures under very cold conditions for a prolonged duration, liver sinusoids can become even more obstructed, particularly after later transplantation with disturbed sinusoidal flow patterns, blood cell stasis, and advanced tissue edema [86]. The clinically applied temperature range between 8 and 11 °C seems optimal and does not lead to higher liver stiffness, as seen with lower temperatures [61]. Equally, reperfusion with oxygen at temperatures below 15 °C is required to protect mitochondria [70,87]. In contrast, reoxygenation after previous ischemia at temperatures beyond 15 °C (Arrhenius breakpoint) triggers higher levels of ROS and lower ATP reloading [70]. Despite such homogenous perfusion parameters applied during HOPE, in contrast to the field of hypothermic kidney perfusion, the impact of pulsatile perfusion (compared to non-pulsatile) is less well assessed [88]. Studies with hypothermic kidney perfusion demonstrated protective effects on ECs conveyed through pulsatile perfusion modes. Vascular pulsatility led to an upregulation of KLF2 and vasodilators, including NO with better EC preservation, compared to non-pulsatile hypothermic perfusion [88].

Such features might become even more important with prolonged perfusion times and should be explored in the future, from the perspective that most clinically utilized devices use a centrifugal pump.

#### 5.1.4. The Role of Perfusion Duration

In addition to perfusion pressures, the best perfusion duration is frequently debated. From a mitochondrial perspective, HOPE/D-HOPE should be performed for at least 90 min to 2 h [69,83,89]. Experimental studies that assessed the mitochondrial metabolism of complex I using real-time spectroscopy showed that perfusate NADH (a marker of complex I function) is metabolized within 90 min to 2 h of HOPE [83]. Thereafter, mitochondria seem to “shut down” and drive low-level maintenance respiration and metabolism, being ready for rewarming and reperfusion [53,69,83]. These results were further paralleled by earlier studies from Germany, demonstrating an ATP recharging within the first and second hours of reoxygenation under cold conditions, while at later time points (3 h), no further ATP level increase was detected [62,89].

Based on logistical challenges in clinical practice, several groups have evaluated the maximal possible or feasible HOPE perfusion duration to move liver transplantation surgery from early hours and nights to daytime. This should, however, be explored with caution, because extended cold perfusion may also trigger injury to the liver SEC cytoskeleton and endoplasmatic reticulum [90]. The reduction of the Disse space was, for example, observed as a result of pressure injury after hypothermic perfusion [90].

Recently, a large multicenter collaboration study demonstrated that clinical HOPE perfusion does not need to be limited to 2 h. With the inclusion of 43 DCD and 50 DBD livers, a median perfusion duration of 4 h 42 min (4:00–8:35 h) and total preservation time of 10 h 50 min (5 h 50 min–20 h 50 min) showed not only the feasibility but also the safety of this prolonged clinical HOPE approach [91]. Boteon et al. also described a prolonged overall preservation and HOPE duration of 11 h 38 min and 5 h 19 min in an extended criteria donor liver with a donor risk index of 2.79 points [92]. This graft was safely transplanted in a medically sick high-MELD recipient with excellent outcomes. Similarly, the group from Berlin reported the use of prolonged HOPE (>4 h) after 12 h 54 min cold storage in an extended criteria donor liver with elevated liver enzymes of 2185 U/L with an additional 20% macrosteatosis [93]. Moreover, the group from Groningen completed a prospective clinical study with prolonged D-HOPE perfusion and liver transplantation, of which the results are eagerly awaited [94]. Endischemic HOPE can therefore be safely extended for up to 24 h for clinical liver transplantation [94]. These clinical results were also supported by an interesting experimental study from the Netherlands. Twenty-four hours of HOPE after 30 min of warm ischemia and cold storage was found to maintain porcine livers’ viability, assessed by ATP reloading, low injury of liver SECs and other cells in histology, and low signs of inflammation. Such livers underwent subsequent normothermic reperfusion for quality assessment, where perfusate and bile chemistry were comparable to livers with short HOPE (Table 1) [95]. The authors added another experimental arm of prolonged HOPE of discarded human DCD livers. HOPE was started after 8–11 h of cold storage and continued for 20 h. Such human livers were found to be viable during subsequent NMP with preserved cellular function in contrast to controls with 24 h of cold storage [95]. The currently available studies point to the safe prolongation of the HOPE approach even with the use of extended criteria donor livers (including DCD grafts) and after relevant cold storage [91,95]. The key perfusion feature is to maintain perfusion pressures low (PV: 3–5 mmHg) to avoid liver SEC injury and prevent mitochondrial activation and other inflammatory signals triggered by too-high shear stress.

**Table 1 ijms-24-10091-t001:** Studies assessing the impact of hypothermic machine perfusion on endothelial cell function and injury in context of liver transplantation.

Author, Year, Country	Number and Species	Liver injury Model, Study Groups	Warm Ischemia	Cold Ischemia before Perfusion	Perfusion Conditions	Active Perfusate Oxygenation	Perfusion Duration	Single/ Dual	Transplantation or NMP	Main Findings	Discussion
Vasuri et al., 2022, Italy [75]	Human livers, *n* = 47	ECD-DBD, HOPE (*n* = 34), SCS only (*n* = 13)	none	6.85 h	Pressure controlled Pressure: PV: 5 mmHg Belzer MP solution (modified UW)	Yes	n.a.	Single (PV)	Transplantation (samples from RCT)	Less EAD in HOPE group (8.8% vs. 30.1% after SCS), LSEC endothelization correlates with HOPE duration	ECD DBD grafts, with transplantation, no real comparison to control group, limited number of markers
Bochimoto et al., 2022, Japan [89]	Porcine, *n* = 6	DCD, HOPE (*n* = 3) vs. HOPE plus rewarming to 22 °C (*n* = 3)	60 min	none (minimal)	Flow controlled, Flow: PV: 0.22 mL/min/g HA: 0.06 mL/min/g, Temperature: 8 °C, Modified UW solution	Yes, pO_2_ 200–300 mmHg	4 h	dual	No	HOPE superior to SCS but inferior to rewarming,	Flow controlled, no cold storage control group, compared to controlled rewarming (8–22 °C), no quantitative endpoint analysis
De Vries et al., 2021, The Netherlands [74]	Porcine (*n* = 6 each group)	DCD livers, HOPE vs. D-HOPE	30 min	SCS duration n.a.	Pressure controlled, Pressure: PV: 5 mmHg, HA: 25 mmHg, Belzer MP solution (modified UW)	Yes, 50–80 kPa (375–600 mmHg).	2 h	Single and dual	NMP (4 h)	No difference between single and dual HOPE, both techniques protect endothelial cells and other liver tissue cells.	No transplantation
Brüggenwirth et al., 2020, The Netherlands [95]	Porcine (*n* = na), human (*n* = 2)	DCD, 2, 6, 24 h (pigs) DHOPE vs. 24 h SCS, human: 20 h HOPE	30 min (pig)	2 h (porcine), 8–11 h (human)	Pressure controlled, Pressure: PV: 5 mmHg, HA: 25 mmHg, Belzer MP solution (modified UW)	Yes, 50–80 kPa (375–600 mmHg).	2, 6, 24 h (porcine), 20 h (human)	dual	NMP (3 or 4 h)	Prolonged D-HOPE is safe and protects endothelial cells and other liver cells, being viable after NMP following 24 h of HOPE	No transplantation, only 2 human livers
Kanazawa et al., 2019, Japan [79]	Porcine livers, *n* = 15	DCD + SCS, HOPE (*n* = 5), HOPE plus rewarming (*n* = 5)	60 min	None (minimal)	Pressure controlled, Pressure: PV: 3–5 mmHg; HA: 30–50 mmHg, Temperature: 8 °C	Yes, pO_2_ 200–300 mmHg	4 h	dual	2 h NMP	Less intra-sinusoidal platelet aggregation, inflammation, more ATP with HOPE	No transplant model but normothermic ex-situ reperfusion
Kron et al., 2018, Switzerland [77]	Human, *n* = 6, rat, *n* > 32	Severe macrosteatosis, HOPE, deoxygenated (HNPE), SCS (*n* = 8 each)	No	12 h	Pressure controlled Pressure: PV: 3 mmHg Belzer MP solution (modified UW)	Yes, 60–80 kPa	2 h	Single (PV)	Transplantation	SEC-protection through HOPE, sign. lower MMP3&8, vWF, ET-1, TIMP, CXCL5&10 expression, better survival, reduced liver injury, better graft survival (human) with HOPE	Only 6 severely steatotic human livers
Burlage et al., 2017, The Netherlands [80]	Discarded human livers, *n* = 18	DCD (*n* = 15), DBD (*n* = 3)	n.a.	7.18 h	Pressure controlled, pressure: PV: 5 mmHg, HA: 25 ± 5 mmHg, Belzer MP solution (modified UW)	Yes, 50–80 kPa (375–600 mmHg).	2 h	dual	6 h NMP	better endothelial cell function through up regulation of mechano-sensitive/cytopro-tective genes, better PV and HA flow, less expression of ET, sign. higher expression of NO, TM, eNOS, HIF-1α, VEGFa, HO-1 with HOPE;	No transplantation but reperfusion and liver evaluation during 6 h of NMP
Op den Dries et al., 2014, The Netherlands [78]	Porcine, *n* = 18	SCS vs. HOPE (*n* = 9 each), DCD	30 min asystolic DWIT	None (minimal)	Pressure controlled Pressure: PV: 5 mmHg, HA: 25 ± 5 mmHg Belzer MP solution (modified UW)	Yes, 100%	2 h	Dual	2 h NMP	HOPE prevents arteriolonecrosis of the peribiliary vascular plexus and protects SEC as shown with less caspase 3 positive SECs	No transplantation, 2 h NMP
Schlegel et al., 2014, Switzerland [83]	Rat, *n* = 64	DCD with SCS and HOPE or NMP with transplantation	30 or 60 min asystolic DWIT	5 h	Pressure controlled Pressure: PV: 3 mmHg Belzer MP solution (modified UW), 8 °C	Yes, >80 kPa	2 h	Single (PV)	Transplantation	SEC protection through HOPE, sign. lower MMP3&8, TIMP, CXCL10 expression, better survival, reduced liver injury, better survival (human)	Rodent model
Schlegel et al., 2013, Switzerland [82]	Porcine, *n*=46	DCD with SCS and HOPE, low (3 mmHg) and high (8 mmHg perfusion pressure, +/−oxygen	60 min	7 h	Pressure controlled Pressure: PV: 3 or 8 mmHg Belzer MP solution (modified UW), 8 °C	Yes, >80 kPa (HOPE group), deoxygenated (HNPE)	2 h	Single (PV)	Reperfusion (NMP)	More injury with high pressure HOPE, particularly liver SECs. High pO_2_ protected mitochondria better with more energy reloading	No transplantation, high injury, different pressures, no dual perfusion as comparator cohort
Henry et al., 2012, USA [76]	Human, *n* = 33	ECD-DBD; SCS (*n* = 15), HMP (*n* = 18)	n.a.	5.1 h	Flow controlled, 0.667 mL flow /g liver/min, pressure: PV: 2.9 ± 0.08 mmHg, HA; 5.5 ± 0.15 mmHg, temperature: 4–8 °C;	Not active	4.3 ± 0.9 h	dual	Transplantation	Tissue samples (PCR) demonstrated HMP-protection including SECs, sign. Less expression of 7 adhesion molecules (seven adhesion molecules: CXCL14, CCL21, CXCL1, ICAM-1, P-Selectin, MCP-1, SDF-1a)	No active oxygenation
T’Hart et al., 2006, The Netherlands [81]	Rat, *n* = 27	Perfusion pressure 2/12.5, 4/25 or 8/50 mmHg (PV, HA), *n* = 3 (1 h HOPE), *n* = 6 each (24 h HOPE)	DBD with SCS or HOPE (different pressures)	None	Pressure controlled Pressure: PV: 2–8; HA: 12.5–50 mmHg, Belzer MP solution	Yes, 100%	1 h and 24 h	Dual	No	PV and HA pressure of 4 and 25 mmHg achieved good cortical perfusion, with minimal endothelial cell death & ATP reloading, higher pressures injured the organ, too low pressures resulted in hypoperfusion	Not transplanted or reperfused, DBD only, HOPE instead, of cold storage, no group with PV perfusion only

Studies with hypothermic or hypothermic oxygenated perfusion assessing the impact on endothelial cell injury or function and perfusion quality within the last 10 years. t’Hart et al. explored the perfusion quality during D-HOPE with different perfusion pressures for PV and HA, defining the best possible conditions for dual HOPE (PV: 4 mmHg, HA 25 mmHg); DBD: Donation after brain death; DCD: Donation after circulatory death; ECD: Extended criteria donors; HA: Hepatic artery; HMP: Hypothermic machine perfusion; HNPE: Deoxygenated HOPE; HOPE: Hypothermic oxygenated perfusion; NMP: Normothermic machine perfusion; MP: Machine perfusion; PV: Portal vein; RCT: Randomized controlled trial; SCS: Static cold storage; TM: Thrombomodulin; n.a.: Not available.

### 5.2. Subnormothermic Machine Perfusion (SNMP)

Although perfusion concepts using a clear (non-blood cell-based) solution with direct oxygenation at hypo- and subnormothermic temperatures were not systematically compared in a comprehensive experimental or clinical study, and the protection of SECs was explored at different temperatures below normothermia. Post et al. found that human umbilical vein endothelial cells remain viable at best under lower temperatures up to 20°C [96].

The following experimental studies explored the effect of SNMP on EC integrity. In 2009, Vairetti et al. compared obese and lean rodent livers that were perfused either in subnormothermic (20 °C) or hypothermic (8 and 4 °C) conditions to evaluate the functional integrity of steatotic livers. Cold-stored livers demonstrated a 2-fold increase in TNF-α and caspase-3 activity compared to steatotic livers preserved by SNMP at 20 °C [97]. A few years later, the same group demonstrated a statistically significant reduction of SEC apoptosis in fatty rodent livers exposed to SNMP compared to cold storage [98].

Other groups explored the effect of SNMP in pre-injured grafts with warm and cold ischemia. Berendsen et al. studied the effect of SNMP (21 °C) on outcomes after rodent liver transplantation (DBD and DCD) [99]. One-month recipient survival rates were 100% and 83.3% in standard grafts and risky livers with warm ischemia (DCD), respectively. In this model, a significant survival benefit was achieved with SNMP compared to cold storage [99].

Next, the Toronto group explored the effect of SNMP at slightly higher temperatures (33 °C) on DCD porcine livers in a transplant model. Posttransplant outcomes were superior with lower EC injury and bile duct necrosis compared to cold storage [100]. Using their transplant protocol with SNMP, the Toronto group demonstrated encouraging results with standard livers showing a reduction of EC and bile duct injury. Of interest is that the perfusion improved outcomes even compared to grafts with minimal cold storage time [101]. In the next step, the same group added anti-inflammatory molecules to the perfusate (including alprostadil, n-acetylcysteine, carbon monoxide, and sevoflurane), with a further reduction of EC injury [102].

Such findings with DCD livers were further paralleled by studies from Japan. Subnormothermic perfusion (20–25 °C) reduced apoptosis and preserved the parenchymal structure of DCD rodent livers [103]. Healthier SECs and better-preserved liver microvasculature were found after SNMP as demonstrated by scanning electron microscopy [103]. Next, a significant reduction of transaminases and mitochondrial glutamate dehydrogenase release was observed in fatty livers treated with SNMP as shown by Okamura et al. [104]. This study also demonstrated the protection of mitochondria, reduced inflammation with lower HMGB-1 expression, and an overall healthier microvascular structure including SECs [104]. Another Japanese group demonstrated the feasibility to split livers during SNMP using an experimental porcine model [105].

Next, Bruinsma et al. perfused seven discarded human livers with SNMP (21 °C) and showed good liver function [64]. In their second project, the same group described their experience with 21 human livers (DBD = 3, DCD = 18) with SNMP. Significantly increased tissue ATP content was observed in DCD livers with 30 min of warm ischemia exposed to 3 hours of SNMP. Mitochondrial protection was further demonstrated with minimal swelling and retained discernable inner and outer membranes and cristae assessed with electron microscopy [106].

A few experimental studies compared the effect of SNMP with hypothermic perfusion. While Furukori et al. found only marginal benefit of both subnormo- and hypothermic perfusion in DCD livers with extended warm ischemia (60 min), some benefits were shown compared to cold storage alone [107]. Bochimoto et al. showed the protective HOPE effect in porcine DCD livers with 60 min warm ischemia time. The authors described, however, better protection of the microvasculature by SNMP [90]. The reason behind these results could be discussed with the perfusion conditions used here, which was in contrast to most other studies. The perfusion system was flow- (and not pressure) controlled with portal vein flow of 0.22 mL/min/g (hepatic artery: 0.06 mL/min/g). Sadly, the study was low in the number of experiments (*n* = 3), and the endpoints were not quantitatively analyzed [90]. Liver SEC damage can be easily induced with perfusion (or even flush) in too-cold conditions using high perfusion pressures that are maintained for prolonged times [82]. Better mitochondrial protection at colder temperatures should be combined with the benefit of slightly higher perfusion temperatures observed in liver SECs. The currently available perfusion devices for clinical HOPE cannot properly decrease the temperature to 4 °C, which appears beneficial for the preservation of liver SECs. Clinical HOPE at 8–12 °C likely protects the microvasculature best, considering the use of a pressure-controlled system with 3–5 mmHg PV pressures [61,62,82]. SNMP studies are summarized in Table 2.

### 5.3. Normothermic Machine Perfusion (NMP)

The role of NMP should be analyzed in the context of the different modalities applied. This warm perfusion technique can be considered organ preservation technology, replacing significant parts of cold ischemia, with expected protective effects on liver SECs or, secondly, as a liver perfusion technique often used endischemically, i.e., after the relevant cold storage and liver transport (Table 3). This second approach, though more practical and easier logistically, conveys a higher risk of developing elevated levels of IRI injury affecting the microcirculation and inducing chronic inflammation in the organ’s periphery. Such findings are supported by the study by Nassar et al., who demonstrated in DCD porcine livers the benefit of the additional use of vasodilators during endischemic NMP, with lower AST, ALD, and LDH and higher bile production in prostacyclin-perfused livers [108].

**Table 2 ijms-24-10091-t002:** Studies assessing the impact of subnormothermic machine perfusion on endothelial cell function and injury.

Author, Year, Country	Number & Species	Liver Injury Model, Study Groups	Warm Ischemia	Cold Ischemia before Perfusion	Perfusion Conditions	Active Perfusate Oxygenation	Perfusion Duration	Single/ Dual	Transplantation	Main Findings	Discussion
Kakizaki, 2018, Japan [103]	Rat, *n* = 20	DCD perfused with SNMP vs. DBD	30 min	6 h	23–26 °C	Oxygen pressure inflow: 500–550 mmHg, outflow 150 mmHg.	15/30, 60, 90 min	Dual	No	Improved portal flow, bile production, decreased liver enzymes, cytokines, and increased ATP. SNMP alleviated liver SECs and hepatic microvasculature.	No transplant model, no other perfusion technique as comparison
Okamura 2017, Japan [104]	Rat	Induced fatty DBD	None	10 min	22.5 °C, pressure PV < 3 mmHg, HA 47 mmHg	Yes, pO_2_ 150 mmHg	4 h	Dual	No	Less release of ALT, mitochondrial glutamate dehydrogenase. ATP, bile production, HMGB1, lipid peroxidation, and glutathione significantly improved by SNMP. Electron microscopy showed improved sinusoidal microvasculature and hepatocellular mitochondria.	No transplant model, no other perfusion technique as comparison
Furukori, 2016, Japan [107]	Pigs, *n* = 8	DCD with HMP vs. SNMP	60 min	None	23 °C	Yes	4 h	Dual	No	Lower tissue IL-1b and IL-6 expression with SNMP.	No transplant model, no comparison to other perfusion technique
Goldaracena, 2016, Canada [102]	Pig, *n* = 15	DBD with SNMP vs. NMP vs. SCS	None	2 h	33 °C, pressure HA 60 mmHg, PV 2–4 mmHg	Yes	4 h	Dual	Yes	SNMP reduced IL-6, TNF-α, and galactosidase levels and increased IL-10 levels. After LT, SNMP had lower AST and bilirubin levels. Decreased hyaluronic acid, as a marker of improved endothelial cell function.	Addition of anti-inflammatory strategies improves livers during NMP, transplant model, porcine livers
Spetzel, 2016, Canada [101]	Pigs, *n* = 16	DBD with SCS or SNMP	None	3 h	33 °C, HA 60 mmHg, PV pressure 2–4 mmHg	Yes	3 h	Dual	Yes	Decreased AST, ALP and bilirubin. Improved hyaluronic acid, less apoptosis at histology	Transplant model, porcine livers, no other perfusion technique as comparison
Okada, 2015, Japan [105]	Pig, *n* = 6	DBD splitting with SNMP	None	100 min	20 °C, pressure PV: 5–10 mmHg, HA: 20–30 mmHg	Yes, 160 mmHg	100 min	Dual	Yes	No differences between groups	Liver splitting is feasible during SNMP
Bruinsma, 2016, US [106]	Human, *n* = 21	Discarded human DCD/DBD	Variable	Variable	21 °C, pressure PV: 4–7 mmHg, HA: 50–80 mmHg	Yes, partial O_2_ pressure >700 mmHg	3 h	Dual	No	Improvement of energetic cofactors and redox and reversal of ischemia-induced alterations, including lactate metabolism and increased TCA cycle intermediates	SNMP can assess liver viability
Knaak, 2014, Canada [100]	Pigs, *n* = 10	DCD with SCS vs. SNMP	45 min	4 h	33 °C, pressure: HA: 60 mmHg, PV: 4–8 mmHg	Yes	3 h	Dual	Yes	Decreased ALP and bilirubin, and no signs of bile duct necrosis	Transplant model, no other perfusion technique as comparison
Bruinsma, 2014, US [64]	Human, *n* = 7	Discarded human DCD	28 (23–34) min	685 (473–871) min	21 °C, pressure PV: 4–7 mmHg, HA: 50–80 mmHg	Yes, partial O_2_ pressure >700 mmHg	3 h	Dual	No	Biochemical and microscopic assessment revealed minimal injury during perfusion.	No transplant model, discarded human livers,
Berendsen, 2012, US [99]	Rat, *n* = 20	DBD/DCD with SNMP vs. SCS	60 min	None	21 °C, portal resistance 50&100 mMH_2_O	pO_2_ n.a.	3 h	Dual	Yes	One-month survival was 100% in the Fresh-SNMP and UW-Control groups, 83.3% in the WI-SNMP group and 0% in the WI group	Transplant study, short cold storage, no other perfusion technique as comparison
Boncompagni, 2011, Italy [98]	Rat	Fatty DBD with SNMP + NMP	None	None	20 °C	Yes, O_2_ concentration 550–650 mmHg	6 h SNMP, 2 h NMP	Dual	No	Reduction of sinusoidal cell apoptosis	No transplant model, Steatotic grafts, no other perfusion technique as comparison
Vairetti et al., 2009 [97]	Rat	Fatty DBD with SNMP + NMP	None	None	20 °C	Yes, O_2_ concentration 550–650 mmHg	6 h SNMP, 2 h NMP	Dual	No	Higher ATP/ADP-ratio and bile production. Lower oxidative stress and biliary enzyme release with SNMP	No transplant model, Steatotic grafts, no other perfusion technique as comparison

Fondevilla et al. analyzed 36 DCD porcine livers (18 donor–recipient pairs), which, after a 90 min warm ischemic time (WIT), were preserved either with SCS alone for 4 h, normothermic extracorporeal machine oxygenation (NECMO) for 60 min plus 4 h of SCS, and NECMO for 60 min plus 4 h of NMP [109]. The authors found that 5-day survival rates after transplantation did not vary statistically between the NECMO + CS (83%) and NECMO + NMP (100%); survival was however significantly lower in the cold storage control group (0%). In addition, after transplantation, the levels of transaminases and bilirubin rose in the SCS group while the NECMO + SCS group had a limited increase. Marginally and relatively stable peak levels were noted in the NECMO + NMP group [109]. Looking at the biopsy and inflammatory response, the authors found a lower TNF /IL-6 mRNA expression in the NECMO + NMP group together with lower liver SEC injury [109].

The Groningen group utilized NMP for 6 h after SCS to assess the viability of four discarded human DCD livers [110]. Perfusate and bile samples were analyzed for gamma-GT and LDH levels, biomarkers of biliary epithelial cell injury. In addition, the bile bilirubin concentration was measured as a biomarker of hepatocellular secretory function. There was an initial increase in lactate concentrations, which subsequently decreased to normal values, and bile production was noted throughout the entire perfusion. Bile gamma-GT and LDH concentrations decreased after an initial peak at 90 min of perfusion. At histology, no signs of biliary injury or sinusoidal damage were observed [110].

In 2016, Banan et al. presented an interesting study, where 18 porcine DBD livers were randomly assigned to 6 groups: (1) SCS < 15 min with immediate NMP for 8 h group; (2) 4 h SCS with 4 h NMP; and (3) 4 h SCS with gradual rewarming followed by 4 h NMP [111]. The gradual rewarming to 38 °C was performed over different time periods: 120, 60, 30, and 20 min. Livers that underwent immediate NMP had lower transaminases and b-galactosidase, less SEC damage measured by hyaluronic acid, higher bile production and less biliary epithelial injury (ALP and LDH), and fewer signs of necrosis in histology. Interestingly, the 120 min rewarming group had the lowest overall enzyme levels among all, indicating that gradual rewarming of a cold-preserved liver significantly reduces injury in contrast to a direct start of NMP [111]. Such findings are in parallel to what was described above for HOPE techniques and favor an immediate start of NMP replacing cold storage or lowering the temperature at reperfusion to protect the mitochondria and microvascular environment, thereby paralleling the findings of Nassar et al. (Table 3) [108].

Next, Zhang et al. explored if NMP is superior to SCS using reduced-size DBD livers in a porcine model of liver transplantation [112]. NMP was started directly after the aortic clamping without any cold flush or storage. The authors found no cell necrosis or hepatic sinusoidal and EC damage with NMP. Reduced levels of transaminases, lactate, and LDH were observed after the transplantation of livers, which underwent a split during NMP. This study showed the feasibility of splitting liver grafts during NMP, which appears superior when replacing cold storage [112].

Nostedt et al. from Alberta, Canada, studied 24 porcine DCD livers with the aim to evaluate whether avoidance of the initial cold flush in NMP could potentially avoid the IRI injury [113]. After 60 min of donor warm ischemia time, the livers were flushed either with the Histidine-tryptophan-ketoglutarate (HTK) solution at 4 °C, 25 °C, and 35 °C or an adenosine-lidocaine crystalloid solution at 4 °C, 25 °C, and 35 °C. During NMP, no differences were demonstrated in perfusate lactate and transaminases levels. Moreover, the HTK 4 °C group had the least edema and sinusoidal dilatation, and histologic injury and TNF-α levels were lowest in NMP with 4 °C flush groups. The authors concluded that omitting the cold flush did not show benefits in their DCD model [113].

Recently, Boteon et al. showed the results of an interesting study, where 10 discarded human livers were divided into two groups with either 2 h D-HOPE with sequentially controlled rewarming (COR) and NMP or immediate NMP after D-HOPE omitting the COR phase [114]. A hemoglobin-based oxygen carrier (HBOC)-based perfusate was used in the COR group; the D-HOPE perfusate was discarded and exchanged for HBOC in the other group. The expression of tissue markers representing oxidative injury and the activation of inflammatory cells were comparable in both groups at the start of the perfusion. During the 6 hours of perfusion, there was an overall decrease in tissue expression of injury markers (UCP2 and 4-HNE). Next, CD14 expression levels decreased more in the HOPE + NMP group. Parameters of liver SEC activation, i.e., CD11b and VCAM-1, decreased in a comparable way between the groups. The authors concluded that the uninterrupted combined protocol of HMP + NMP using an HBOC-based perfusate was similar to the interrupted one [114].

In 2020, a Chinese group investigated the role of bone marrow mesenchymal stem cells (BMMSCs) to regulate the immune response during NMP in 20 DCD rodent livers [115]. After 30 min of donor warm ischemia time, the livers were divided into three groups, SCS, NMP, and NMP with BMMSCs. Interestingly, the addition of BMMSCs mitigated mitochondrial damage with limited cell swelling assessed by electron microscopy. In addition, the authors found lower transaminases, more bile production, and better lactate clearance with BMMSCs. Looking at the microcirculation, quantitative analysis showed that the NMP alone and BMMSCs groups had lower CD14 and CD68 expression, suggesting a reduced macrophage activation, compared to cold storage. Markers of liver SEC activation, i.e., ICAM-1 and VCAM-1, were higher in the SCS group. These interesting results demonstrated that the addition of BMMSCs could be of interest to be further explored beyond this rodent model (Table 3) [115].

Recently, the Zurich group developed an enhanced perfusion technology to preserve porcine and human livers for 7 days with NMP in an experimental setting [116]. During the perfusion, cytochrome C, an established marker for mitochondrial membrane injury, did not increase over 7 days and the histological assessment did not reveal signs of relevant necrosis or features of endothelial activation [116]. Human livers, which were discarded after viability testing during HOPE, underwent prolonged NMP [69]. Six livers (6/10) had less injury and inflammation with lower transaminases, cytokines, and uric acid compared to the remaining four (40%) [116]. These six healthier livers demonstrated intact endothelial cells with low levels of Willebrand factor and intercellular-adhesion-molecule 1 in histology. This first report of such prolonged normothermic perfusion was recently paralleled by the group from Australia increasing the perfusion duration further to >12 days [117]. Authors from Zurich also described the need for additional vasodilators to maintain an appropriate flow in the HA and reduce vasoconstriction, a response to higher oxygen levels in the portal vein [118]. Lately, the first case of a successful human liver transplantation after 3 days of NMP was reported with a 1-year follow-up [119]. Based on the aim to develop a model of ex situ liver regeneration, the team from Zurich also explored the feasibility and effect of NMP on partial porcine and human livers. Structural integrity and cellular energy were maintained during prolonged NMP in 21 human hemi-livers [120].

**Table 3 ijms-24-10091-t003:** Studies assessing the impact of normothermic machine perfusion on endothelial cell function and injury.

Author, Year, Country	N and Species	Liver Injury Model, Study Groups	Warm Ischemia	Cold Ischemia before Perfusion	Perfusion Conditions	Active Perfusate Oxygenation	Perfusion Duration	Single/ Dual	Transplantation	Main Findings	Discussion
Yang, 2020, China [115]	Rat, *n* = 20	DCD + SCS or DCD + NMP or DCD + BMMSCs + NMP	30 min	4–8 h, none in the NMP group	PV 2 mL/g/min with 10–12 mm H_2_O pressure	Yes, not specified	8 h	Single (PV)	No	Lower AST, ALT and ALP, more bile production and lactate clearance with BMMSCs + NMP; BMMScs+ NMP led to less apoptosis and mitochondrial damage with low AST levels, less nucleic swelling and mitochondrial oedema	Rodent model, no transplantation
Boteon, 2019, UK [114]	Discarded human livers, *n* = 10	DBD/DCD + SCS and HOPE + NMP vs. HBOC + COR + NMP	8 (7–31) for the DCD	11 h	HOPE PV flow 0.1 mL/min/g NMP flow: PV: 0.25 mL/min/g; HA: 0.75 mL/min/g, NMP pressure HA 30–50 mmHg, PV 8–10 mmHg.	Yes, pO_2_ 80–100 kPa (HOPE); pO_2_: 40 kPa for NMP	2 h HOPE 1 h COR	Dual HOPEDual NMP	No	Downstream activation of the inflammatory cascade and less expression of CD14 in HOPE + NMP compared to cold-to-warm, decrease in CD11b in neutrophil of both groups	Discarded human livers, without transplantation, comparison of different perfusion techniques
Nostedt, 2019, Canada [113]	Pig, *n* = 24	DCD with initial flush either AD or HTK at different temp. with NMP	60 min	None	Pressure HA: 60 mmHg, PV weight-adjusted	Yes, not specified	12 h	Dual	No	Histologic injury and TNF-α levels were lowest in NMP with 4 °C flush groups, no differences in lactate and biochemistry. HTK 4 °C group had the least oedema and sinusoidal dilatation.	Pigs model without transplantation.
Zhang, 2016, China [112]	Mini pigs, *n* = 24	DBD SCS vs. NMP	None	No SCS in NMP group, SCS group 133.8 ± 21.90 min	PV flow 320–580 mL/min with pressure 8–10 mmHg; HA Flow 129–239 mL/min HA pressure 85–100 mmHg	Yes, pO_2_: 80–100 mmHg CO_2_ partial pressure 30–50 mmHg	127.3 ± 37.12 min	Dual	Yes	No cell necrosis or hepatic sinusoids and endothelial cell damage with NMP. Less AST, ALT, lactates and LDH in NMP in postoperative days 3 and 5 (*p* < 0.05).	Pig model, transplant model
Fondevila, 2011, Spain [109]	Pig, *n* = 36	DCD with SCS, NECMO + SCS, NECMO + NMP	90 min	Directly perfused after SCS	SCS: 1L UW 60 min NECMO, 1.8 L/min, 37 °C. NMP: PV: 8 mmHg, HA: 60/40 mmHg, 35.5–37.5 °C.	Yes in NECMO 1.8 L/min NMP: pO_2_: 60/40 mmHg *	60 min NECMO 4 h NMP	Dual	Yes	Lower TNF /IL-6 mRNA expression in NECMO + NMP after reperfusion. Lower endothelial cell injury with NECMO + NMP	Transplant model
Banan, 2016, US [111]	Pig, *n* = 18	DBD with immediate NMP, or with lower temp, 20 or 30 or 60 or 120 min rewarming	None	<15 min in the immediate NMP, 4 h in other groups	Flow 0.45 ± 0.15 L/min HA: and 1.3 ± 0.2 L/min (PV) arterial pressures of 80 ± 5 mmHg and portal pressures 6 ± 1 mm Hg	Yes, 95% O_2_ and 5% CO_2_	8 h in immediate NMP 4 h in other groups	Dual	No	Immediate NMP had lower transaminases, b-galactosidase, less SEC damage measured with hyaluronic acid, less necrosis at biopsy	Immediate NMP vs. 20,30,60, 120 min rewarming, no transplant model.
Op den Dries, 2013, Netherlands [110]	Discarded human livers, *n* = 4	DCD with SCS, perfused with NMP	38 ± 11.9 min §	7 ± 1 h #	Mean pressure HA: 50 mmHg, PV 11 mmHg. Mean HA flow of 283 ± 29 mL/min; mean PV flow: 686 ± 25 mL/min	Yes, 60 kPa	6 h	Dual	No	ALT, gamma-GT remained stable. Lactate decreased, normal pH, no signs of sinusoidal damage. No evidence of apoptosis. Partial loss of the biliary epithelial cell layer	Discarded human livers, no transplant model
Liu, 2013, US [108]	Pig, *n* = 15	DCD with SCS + NMP Prostacyclin or adenosine or none	60 min	Directly NMP after warm ischemia	Mean pressure HA: 90–100 mmHg, PV 7–12 mmHg	Yes *	10 h	Dual	No	Lower AST, ALD and LDH in prostacyclin perfused together with higher bile production. Vasodilatation is key factor for better outcomes with NMP	Pig models without transplantation.
Fondevilla, 2011, Spain [109]	Pig, *n* = 36	DCD with SCS, NECMO + SCS, NECMO + NMP	90 min	Directly perfused after SCS	SCS: 1 L UW 60 min of NECMO, 1.8 L/min, 37 °C. NMP: PV: 8 mmHg, HA: 60/40 mmHg, 35.5–37.5 °C.	Yes in NECMO 1.8 L/min NMP: pO_2_: 60/40 mmHg *	60 min NECMO 4 h NMP	Dual	Yes	Lower TNF /IL-6 mRNA expression in NECMO+ NMP after reperfusion. Lower endothelial cell injury with NECMO + NMP	Pig Model with transplantation

Studies with normothermic machine perfusion assessing the impact on endothelial cell injury or function and perfusion quality. * No kPA available; § is the mean SD of the sum of withdrawal and cardiac death and cold perfusion time; # is the mean SD of the sum (=415 ± 58 min). DBD: Donation after brain death; DCD: Donation after circulatory death; ECD: Extended criteria donors; HMP: Hypothermic machine perfusion; HOPE: Hypothermic oxygenated perfusion; NMP: Normothermic machine perfusion; MP: Machine perfusion; NECMO: Normothermic extracorporeal machine oxygenation; RCT: Randomized controlled trial; SCS: Static cold storage; TM: Thrombomodulin; n.a.: Not available.

## 6. Summary and Future Perspective

Healthy liver endothelial cells are key contributors to maintaining various liver functions and their protection should be in focus when developing novel preservation strategies [32]. Liver SEC dysfunction leads to ongoing inflammation, fibrosis, and increased stiffness with impaired perfusion quality, determining the outcome of liver injury. This has vast implications for the development of the best possible liver preservation technique and immediate and long-term results after liver transplantation, also due to their role to modulate the immune response and tumor recurrence [121]. Although many experimental studies with different machine perfusion concepts are available, most studies are purely descriptive, and a systematic comparison with a focus on the effect on liver SEC is lacking. Many other parameters are essential to developing the best possible ex situ liver perfusion system. Most perfusion conditions are, however, transferred from clinical scenarios or based on the perfusion of other organs, such as kidneys, instead of being systematically developed. While only two studies compared different perfusion pressures during HOPE, this parameter is only one example of a lack of studies, including the role of different temperatures. Further parameters with an effect on endothelial cells including perfusate oxygen levels and perfusate composition should be explored in the context of different levels of organ injury. Moreover, the role of pulsatile and non-pulsatile perfusion modes remains underexplored in the context of liver perfusion. Particularly steatotic livers benefit from a well-preserved microvasculature to avoid secondary hypoxia due to sinusoidal obstruction. Perfusion conditions and timing should be compared systematically in the context of the upcoming concept of perfusion hubs, where organs may undergo cold storage before and after machine perfusion [122,123]. The exposure of activated and proinflammatory liver SECs to additional cold ischemia for transport after NMP or HOPE should be explored systematically to determine the best possible timing. More prospective clinical comparison studies (i.e., with endischemic HOPE vs. NMP, NCT04644744) are of interest and should ideally include the assessment of liver tissues to compare the quality of mitochondrial and SEC protection. New interventions and molecules added to perfusates in the future may only achieve their full potential to improve and maintain livers during long-term perfusion when the best possible perfusion conditions are used.

## Figures and Tables

**Figure 1 ijms-24-10091-f001:**
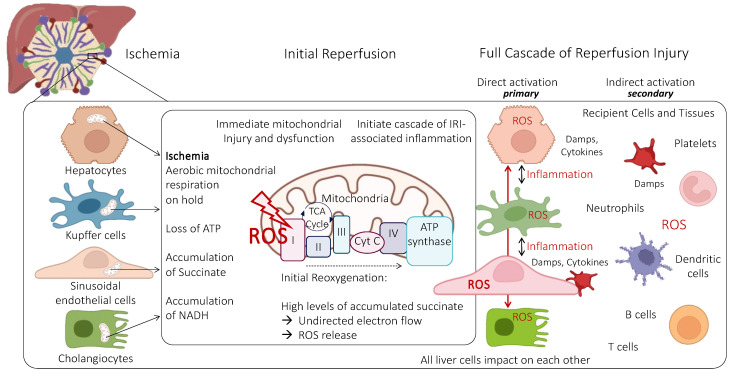
Role of mitochondria in various liver cells during ischemia-reperfusion injury (IRI). All types of liver cells have different metabolic capacity and subsequent mitochondria. Although hepatocytes contain the highest number of mitochondria, this subcellular powerhouse plays an equally key role in all other cells. With prolonged ischemia, the electron transfer is on hold and the TCA cycle and other metabolic processes shift to anaerobic features, accumulating potentially detrimental metabolites, such as succinate. When oxygen is reintroduced, high succinate levels trigger an uncoordinated and retrograde electron flow with the release of reactive oxygen species (ROS) from all affected mitochondria. This initial trigger is the key instigator of the entire IRI cascade with direct damage to affected cells and indirect injury of neighboring cells initially not severely injured. The activated microenvironment of a newly implanted liver in turn attracts circulating cells in the recipient’s blood, including neutrophils and platelets, which attach to SECs. ATP: Adenosine-trisphosphate; Cyt C: Cytochrome C; DAMPs: Danger-associated molecular patterns; ROS: Reactive oxygen species; SECs: Sinusoidal endothelial cells; TCA-cycle: Tricarboxylic acid cycle.

**Figure 2 ijms-24-10091-f002:**
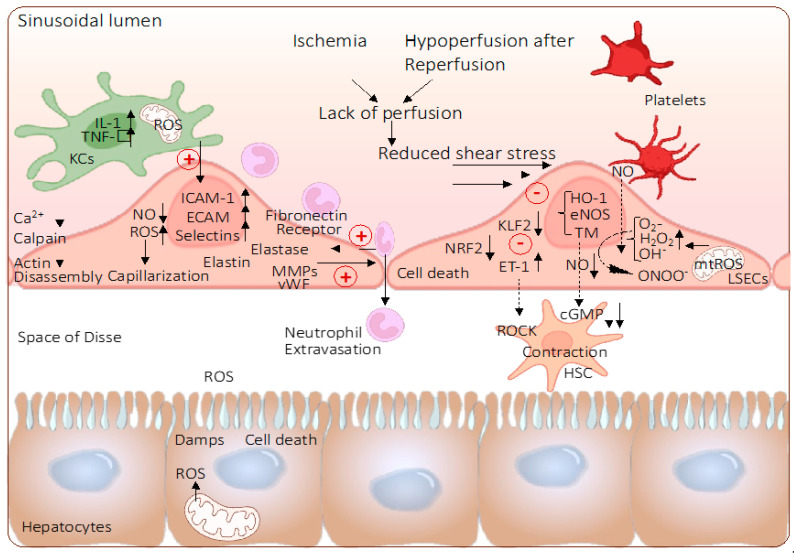
Proinflammatory liver microenvironment during reperfusion. The initial injury is triggered by mitochondria, either inside the corresponding liver SEC or by ROS molecules released from other cells. These instigator molecules are the first domino stope to initiate the IRI cascade. SECs are impaired in function through two mechanisms, initially by their own IRI and in a second wave once these cells are situated peripherally to initially damaged liver SECs that inflame their microenvironment with attachment of neutrophils and platelets leading to second hypoxia with an ongoing cascade of inflammation. Key mediators are mtROS, KLF2, NRF2, NO triggering the expression of ICAM and ECAM, and other cytokines interacting with macrophages and other recipient cells circulating through the microenvironment of the newly implanted liver. Various levels of liver SEC shear stress affect the expression of vasoconstrictive and vasodilative molecules (i.e., ET-1) that communicate with hepatic stellate cells (HSC) in the space of Disse. ECAM: Endothelial cell adhesion molecule; ICAM: Intercellular adhesion molecule; IRI: Ischemia reperfusion injury; KLF2: Kruppel-like factor 2; NO: Nitric oxide; NRF2: Nuclear factor erythroid 2–related factor 2 (regulator of cellular resistance towards oxidative stress); mtROS: Mitochondrial reactive oxygen species.

**Figure 3 ijms-24-10091-f003:**
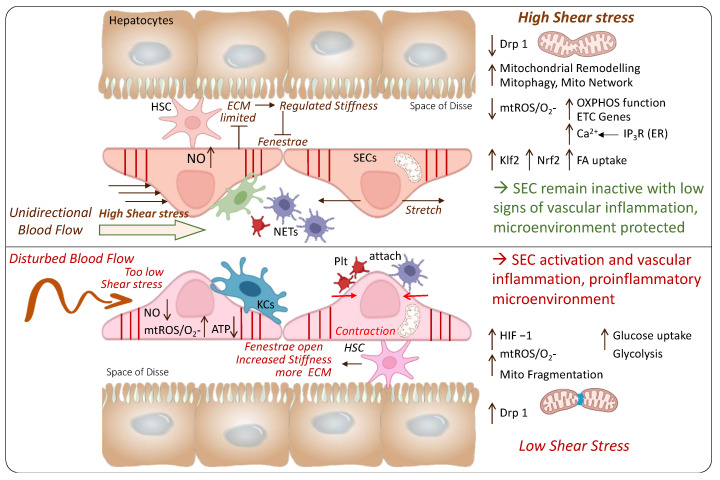
Link between blood flow, shear stress, and mitochondrial dysfunction. Different pathways of liver SEC injury are shown. Disturbed shear stress leads to mitochondrial injury in SECs, exposing hepatocytes and other parenchymal cells to additional inflammation and injury through a lack of proper SEC function and protection. During ischemia−reperfusion injury, mitochondria in other cells (i.e., hepatocytes and macrophages) also release ROS and DAMPs and promote the maintenance of inflammation in the microvascular environment. The ongoing inflammation leads to a high risk of further disturbed sinusoidal flows and obstruction with secondary hypoxia.

## Data Availability

The data presented in this study are openly available.

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
