# Peer review of "Endothelial Cells and Mitochondria: Two Key Players in Liver Transplantation"

_ijms, 2023, doi:10.3390/ijms241210091_

Round 1

Reviewer 1 Report

The manuscript of “Endothelial Cells and Mitochondria: Two Key Players in Liver Transplantation” by A. Parente and co-authors aims to review the relationship between blood flow, shear stress and mitochondrial dysfunction in liver sinusoidal endothelial cells (LSECs) and other liver cells during ischemia-reperfusion injury (IRI), which may be associated with liver transplantation. The authors have discussed the role of mitochondria in various liver cells and the pathways of LSEC dysfunction during different IRI-phases in context of liver transplantation. Special attention was paid to currently ex-situ perfusion techniques and their impact on mitochondrial function and their potential to protect liver SECs. The authors have summarized studies assessing the impact of hypothermic, subnormothermic, and normothermic machine perfusion on endothelial cell function and injury in context of liver transplantation.

The manuscript is quite interesting and well structured; all the conclusions are supported by the data obtained. The review is well illustrated (contains 3 schematic illustrations) and includes 123 literature sources, most of which have been published in the last five years. The manuscript contributes to the systematization of current knowledge about the role of liver SEC dysfunction in the development of inflammation, fibrosis and increased stiffness with impaired perfusion quality, determining the outcome of liver injury.

Minor editing of English language required.

Author Response

Reviewer 1:

The manuscript of “Endothelial Cells and Mitochondria: Two Key Players in Liver Transplantation” by A. Parente and co-authors aims to review the relationship between blood flow, shear stress and mitochondrial dysfunction in liver sinusoidal endothelial cells (LSECs) and other liver cells during ischemia-reperfusion injury (IRI), which may be associated with liver transplantation. The authors have discussed the role of mitochondria in various liver cells and the pathways of LSEC dysfunction during different IRI-phases in context of liver transplantation. Special attention was paid to currently ex-situ perfusion techniques and their impact on mitochondrial function and their potential to protect liver SECs. The authors have summarized studies assessing the impact of hypothermic, subnormothermic, and normothermic machine perfusion on endothelial cell function and injury in context of liver transplantation. 

The manuscript is quite interesting and well structured; all the conclusions are supported by the data obtained. The review is well illustrated (contains 3 schematic illustrations) and includes 123 literature sources, most of which have been published in the last five years. The manuscript contributes to the systematization of current knowledge about the role of liver SEC dysfunction in the development of inflammation, fibrosis and increased stiffness with impaired perfusion quality, determining the outcome of liver injury.

Minor editing of English language required.

Our reply:

We thank the reviewer for the kind evaluation of our manuscript and the positive feedback. The manuscript underwent English language editing.

Reviewer 2 Report

  • This paper aims to describe the role of LSEC and their mitochondrial function in liver transplantation, and it succeeds. It is an extensive review paper on the subject and various perfusion methods. It opens the door for eventually using drugs during perfusion to enhance mitochondria function, which might be the future.
  • Maybe there shouldn't be so many reference citations of works by the authors

Author Response

Reviewer 2:

This paper aims to describe the role of LSEC and their mitochondrial function in liver transplantation, and it succeeds. It is an extensive review paper on the subject and various perfusion methods. It opens the door for eventually using drugs during perfusion to enhance mitochondria function, which might be the future.

Maybe there shouldn't be so many reference citations of works by the authors

Our reply:

We thank the reviewer for the kind evaluation of our manuscript and the comment. We have revised the references accordingly.

Reviewer 3 Report

My specific comments are in a separate file 

In this review, Parente et al. attempted to critically analyze the roles of liver sinusoidal cells and their mitochondria in cardiac transplantation. While the topic appears relatively underexplored and therefore is of interest to experts in the field of liver transplantation, the review appears of limited interes to the broad readership of IJMS.

This reviewer got stuck within the first 100 lines because of poor English, which obscured the scientific content. Therefore, this reviewer’s suggestion is to extensively revise the manuscript for English as well as scientific coherence and logic.

I couldn't get through the review because of the language. English must be improved

Author Response

Reviewer 3:

My specific comments are in a separate file 

In this review, Parente et al. attempted to critically analyze the roles of liver sinusoidal cells and their mitochondria in cardiac transplantation. While the topic appears relatively underexplored and therefore is of interest to experts in the field of liver transplantation, the review appears of limited interes to the broad readership of IJMS.

This reviewer got stuck within the first 100 lines because of poor English, which obscured the scientific content. Therefore, this reviewer’s suggestion is to extensively revise the manuscript for English as well as scientific coherence and logic.

Our reply:

We thank the reviewer for thorough evaluation of our manuscript.

We have received the separate file from the editorial office and have revised the manuscript accordingly. Please see section below.

Line 13 and line 31. ‘Building the inner layer of our blood vessels, endothelial cells form an important barrier protecting deeper parenchymal cells in our organs.‘ Protecting from what? Please, clarify

Our reply:

We thank the reviewer and have revised the sentence accordingly.

Line 14. ‘ Previously considered as passive cell line’ endothelium is not a cell line. Please, correct

Our reply:

This has been corrected.

Line 16. ‘ Comparable to other cells,’ ComparED. Please, correct

Our reply:

This has been corrected.

Line 33. ‘ active key players’ Key players can not be passive. Redundant

Our reply:

This has been corrected.

Line 42. ‘Liver SECs can remain quiet for decades’ It is unclear what quiet means in this context. Is it quiescent? ‘Quiet’ is usually an antonym for ‘loud’

Our reply:

This has been corrected.

Line 43.’ dan-ger associated molecular pattern (damps). Please, capitalize DAMPS

Our reply:

This has been done.

Line 50. ‘A special focus is led on the effect’ What does this mean?

Our reply:

We thank the reviewer and have revised this section.

Line 57’ Mitochondria are increasingly recognized for their fundamental role ... and outcomes after transplantation of solid organs ’ It is hard to understand what is meant here

Our reply:

We thank the reviewer for this comment. The sentence highlights the role of mitochondria in various processes in health and disease and was revised.

Line 62. ‘Of particular importance in endothelial cells is the contribution to required cellular and mitochondrial calcium levels and the homoeostasis.’ Perhaps, homEOstasis? Don’t cellular levels include mitochondrial (as cells include mitochondria)? Perhaps, cytosolic, not cellular? Isn’t the maintenance of cellular calcium levels a part of homeostasis? In that case, this sentence is redundant.

Our reply:

We thank the reviewer and have revised this sentence.

Line 64. ‘mitochondrial membrane potential, retained by complex I, III and IV providing the full OXPHOS-capacity to build ATP.’ Perhaps, ‘generated’, not ‘retained’. Also, “build ATP” sounds weird.

Our reply:

We have revised this sentence.

Line 70. ‘The various metabolic features observed in endothelial cells gain more recognition because of the limited success seen with growth-factor targeted therapies alone ‘This sentence makes no sense

Our reply:

We thank the reviewer and have revised this sentence.

Comments on the Quality of English Language:

I couldn't get through the review because of the language. English must be improved

Our reply:

The manuscript underwent editing of English language to improve the overall readability. In addition, some of the references were revised based on the suggestions from Reviewer 2.

Round 2

Reviewer 3 Report

This revised version of the manuscript is improved. However, it remains difficult to read and understand beyond the first 100 lines, which were specifically criticized in my first review.

For example, lines 103-110: "The various blood flow patterns are additional key contributors, triggering the metabolic switch in ECs. Hong et al recently explored such features in a model of carotid artery ligation1. ECs calm down [ it is unclear what "calm down" means as applied to endothelial cell] or remain quiescent with limited glycolysis and pronounced OXPHOS, features depended on ATP production with unidirectional flow patterns12. [as written, it suggests that glycolysis and OXPHOS depend on ATP production. Since both glycolysis and OXPHOS are ways to produce ATP, this sentence makes no sense] In contrast, disturbed flows initiate an EC switch to pro-inflammatory phenotypes with increased glycolysis and reduced OXPHOS9,13. Based on such recent studies, the augmentation of OXPHOS might be a promising target for new therapeutics to reduce the inflammation and prevent chronic stages [it is unclear what chronic stages are meant here. Also, the conclusion is logically faulty: it implies that reduced OXPHOS causes inflammation. However, the authors present no evidence for such a causal effect. Instead, their data indicate that both inflammation and OXPHOS changes are caused by non-laminar blood flow].

It is difficult to impossible to follow the logic of many statements in this review (see my response to the authors). If this were a literature review for a graduate student's thesis at our institution, the student would have been sent back to rewrite, and a major professor would have been advised to work with a student on the clarity.

Author Response

Reviewer 3:

This revised version of the manuscript is improved. However, it remains difficult to read and understand beyond the first 100 lines, which were specifically criticized in my first review.

Our reply:

We thank the reviewer for the thorough evaluation of our manuscript and the feedback.

For example, lines 103-110: "The various blood flow patterns are additional key contributors, triggering the metabolic switch in ECs. Hong et al recently explored such features in a model of carotid artery ligation1. ECs calm down [ it is unclear what "calm down" means as applied to endothelial cell] or remain quiescent with limited glycolysis and pronounced OXPHOS, features depended on ATP production with unidirectional flow patterns12. [as written, it suggests that glycolysis and OXPHOS depend on ATP production. Since both glycolysis and OXPHOS are ways to produce ATP, this sentence makes no sense]  In contrast, disturbed flows initiate an EC switch to pro-inflammatory phenotypes with increased glycolysis and reduced OXPHOS9,13. Based on such recent studies, the augmentation of OXPHOS might be a promising target for new therapeutics to reduce the inflammation and prevent chronic stages [it is unclear what chronic stages are meant here. Also, the conclusion is logically faulty: it implies that reduced OXPHOS causes inflammation. However, the authors present no evidence for such a causal effect. Instead, their data indicate that both inflammation and OXPHOS changes are caused by non-laminar blood flow].

Our reply:

We thank the reviewer for the comments and the thorough assessment of our manuscript. Hong et al in their study demonstrated that inflammation and reduced OXPHOS are caused by disturbed flow. In vessels where EC were exposure to unidirectional flow more elongated mitochondria were present and OXPHOS was enhanced in ECs. Therefore, the protection of OXPHOS could be of interest to reduce endothelial cell activation and inflammation. We have modified the text accordingly. Please see page 2-3 line 84 and beyond.

Excerpt:

“The mitochondrial metabolism was described to contribute to the regulation of the vessel diameter. The various blood flow patterns play an additional key role, triggering a metabolic switch in ECs. Hong et al recently explored such features in a model of carotid artery ligation [1]. The authors demonstrated that mitochondrial fragmentation is increased in EC's exposed to disturbed flows. The significant induction of mitochondrial fragmentation was associated with EC activation. In contrast, elongated mitochondria were predominant in EC's exposed to unidirectional flow [1], a flow pattern found to decrease mitochondrial fragmentation, improve fatty acid uptake and OXPHOS [1]. Conversely, disturbed flows trigger EC activation and switch to pro-inflammatory phenotypes with increased glycolysis and reduced OXPHOS [9, 13]. Based on such recent studies, the augmentation of OXPHOS might be a promising target for new therapeutics to reduce the acute inflammatory response and potentially prevent ongoing inflammation”.